# Systematic mapping of contact sites reveals tethers and a function for the peroxisome-mitochondria contact

Nadav Shai[1], Eden Yifrach[1], Carlo W.T. van Roermund[2], Nir Cohen [1], Chen Bibi[1], Lodewijk IJlst[2], Laetitia Cavellini[3], Julie Meurisse[3], Ramona Schuster[4], Lior Zada[1], Muriel C. Mari[5], Fulvio M. Reggiori[5], Adam L. Hughes[6], Mafalda Escobar-Henriques[4], Mickael M. Cohen[3], Hans R. Waterham[2], Ronald J.A. Wanders[2], Maya Schuldiner [1] & Einat Zalckvar[1]

The understanding that organelles are not floating in the cytosol, but rather held in an organized yet dynamic interplay through membrane contact sites, is altering the way we grasp cell biological phenomena. However, we still have not identified the entire repertoire of contact sites, their tethering molecules and functions. To systematically characterize contact sites and their tethering molecules here we employ a proximity detection method based on split fluorophores and discover four potential new yeast contact sites. We then focus on a little-studied yet highly disease-relevant contact, the Peroxisome-Mitochondria (PerMit) proximity, and uncover and characterize two tether proteins: Fzo1 and Pex34. We genetically expand the PerMit contact site and demonstrate a physiological function in β-oxidation of fatty acids. Our work showcases how systematic analysis of contact site machinery and functions can deepen our understanding of these structures in health and disease.

[1] Department of Molecular Genetics, Weizmann Institute of Science, Rehovot 7610001, Israel. [2] Laboratory Genetic Metabolic Diseases, Academic Medical Center, University of Amsterdam, Amsterdam 1105 AZ, The Netherlands. [3] Laboratoire de Biologie Moléculaire et Cellulaire des Eucaryotes, Institut de Biologie Physico-Chimique, Sorbonne Universités, UPMC Univ Paris 06, CNRS, UMR8226, 75005 Paris, France. [4] Institute for Genetics, CECAD Research Center, University of Cologne, 50931 Cologne, Germany. [5] Department of Cell Biology, University of Groningen, University Medical Center Groningen, Amsterdam, The Netherlands. [6] Department of Biochemistry, University of Utah School of Medicine, Salt Lake City, UT 84112, USA. Correspondence and requests for materials should be addressed to M.S. (email: maya.schuldiner@weizmann.ac.il) or to E.Z. (email: einat.zalckvar@weizmann.ac.il)

Membrane contact sites are cellular domains where membranes of two organelles are kept in close proximity by protein–protein or protein–lipid tether complexes[1,2]. Contact sites are a conserved phenomenon and have multiple fundamental functions including arrangement of the cellular landscape; exchange of lipids, ions, and other small molecules; organelle inheritance and fission[3]. In the last years tethering, effector and regulatory proteins of contact sites have started being identified[1,2]. However, it is clear that only a few pieces of the complete cellular puzzle of contact sites have been assembled thus far and that therefore new contact sites, more functions, regulators and resident proteins await discovery.

We have recently shown that it is possible to detect contact sites, even if they are unknown or their resident proteins are unidentified, by using a synthetic reporter[2]. The approach relies on simple tagging of abundant membrane proteins from two different organelles with half a fluorophore each. Only when the organelles come into extremely close proximity, as can be found in contact sites, will the two halves of the protein be able to traverse the distance, enable the formation of a full fluorophore and report on the contact by a fluorescent signal. This tool has now been used in both the yeast *Saccharomyces cerevisiae* (from here on termed yeast)[2] and mammals[4–6] to explore known contact sites.

With the aim of identifying novel contact sites, we systematically assayed for proximity between pairwise combinations of five cell compartments in yeast and found four potential new contacts never before described: plasma membrane (PM)-vacuole, PM-lipid droplet (LD), PM-peroxisome and peroxisome-vacuole.

Concentrating on the Peroxisome-Mitochondria (PerMit) contact site we performed a high content screen visualizing the PerMit reporter in thousands of overexpression strains and identified two novel tethers, Fzo1 and Pex34, whose overexpression dramatically increased the contact extent and that abide to all of the requirements to be termed tethers[2]. Importantly, revealing the identity of tethers allowed us to genetically expand the contact site and demonstrate, for the first time, a physiological role for the PerMit contact in β-oxidation of fatty acids, a process that requires a tight collaboration between peroxisomes and mitochondria.

The discovery of new contact sites, molecules that transfer through contacts and unappreciated tethering paradigms should broaden the scope of our thinking on contact sites especially in disease models where the inherent role and importance of contact sites are only now starting to be uncovered.

## Results

**Systematic analysis of contact sites.** Many contact sites have been described to date and several have been intensively studied[2]. However, we still do not know the full repertoire of contact sites that exist in cells. To identify and characterize contact sites between organelles in a systematic way, we choose to build on a bimolecular fluorescence complementation assay[7,8] that we and others have previously demonstrated as a powerful tool for visualizing known contact sites[2,6]. Recently this fluorescence complementation assay has also been used to study the endoplasmic reticulum (ER)-PM contact site in yeast[9] and the ER-mitochondria contact site in both yeast[2] and mammals[4–6]. In short, we coated the perimeter of each organelle with half of a Venus fluorescent protein. In areas of a contact site between the two organelles, the interaction of the two halves and the formation of a full fluorophore is enabled (Fig. 1a).

To systematically use the split fluorescence reporter, we first further characterized its properties. To this end, we created

reporters for the nuclear–vacuolar junction (NVJ)[10], the ER-mitochondria contact (MAM)[11] and the contact site between mitochondria and the vacuole (vCLAMP)[12] (Supplementary Fig. 1a). We verified that the reporters co-localize with known contact site residents such as Nvj1[10] and Mdm1[13] with the NVJ reporter (Supplementary Fig. 1b) and Mdm34[11], with the MAM reporter (Supplementary Fig. 1c).

It was previously demonstrated that deletions of a single set of tethers most often do not completely abolish a contact site[2,13–15]. Indeed, we found that deletions of the known MAM or NVJ tethers did not change the MAM nor the NVJ reporter signals (Supplementary Fig. 1d, e, f). However, deletion of six tethering genes (Δtether) shown before to severely affect the ER-PM contact enabled us to track a reduction in contact site extent (Supplementary Fig. 1g)[15]. Inversely, expansion of a contact site was more readily visible. For example, overexpression of Lam6 caused an obvious expansion of the MAM reporter signal (compare Supplementary Fig. 1a and Supplementary Fig. 1h) as expected[14]. In addition, Vam6 overexpression expanded the vCLAMP reporter signal (compare Supplementary Fig. 1a and Supplementary Fig. 1i) in agreement with previous observations[12].

Split Venus has previously been shown to create a stable Venus protein with strong affinity hence creating a non-reversible tether[16]. Indeed our MAM reporter compensated for the loss of the ER–Mitochondria Encounter Structure (ERMES) complex, a well characterized ER-mitochondria tether complex[11] (Supplementary Fig. 1j). Hence the Venus reporter can be used as a powerful tool to easily create synthetic tethers to assay for rescue of defected contacts. However, it demonstrates both the power and the limitations of the split Venus reporter: it enables a snapshot of contact sites, even low abundance or transient ones, but is not dynamic and once created, cannot be eliminated.

For our systematic analysis, we chose 26, highly expressed, membrane proteins of the ER, mitochondria, PM, vacuoles, LD, or peroxisomes (Supplementary Table 1). We first ensured that the candidate proteins are expressed, localized to the right organelle and that their tagged C terminus (′) is facing the cytosol (to enable the formation of the full fluorophore). To do so we tagged the candidate proteins with the C′ part of a Venus fluorophore (VC) while the N′ part of the Venus protein (VN) was expressed as an independent soluble protein in the cytosol (Supplementary Fig. 2a, b). We further continued only with proteins that showed a clear signal and localized to the correct cellular compartment (Supplementary Table 1).

To evaluate the full extent of organelle proximity in the yeast cell, we created pairwise combinations of the reporters. As it is starting to be appreciated[17], we found that all examined organelles interact to some extent (Fig. 1b). Interestingly, the pattern of the reporter correlated with the known shapes of the contacts. For example, the shape of the vacuole-mitochondria contact (vCLAMP)[12] was crescent-like, while the ER-PM contact (PAM) spread on most of the PM area as previously described[18]. Most other reporters gave a punctate signal, however, their number was variable. For example, the LD-ER reporter signal displayed multiple puncta, suggesting that most LDs are in contact with the ER, while only few puncta of LD-mitochondria or LD-vacuole signals were observed, suggesting that only subpopulations of LDs create contacts with these latter organelles[19].

To verify that the newly identified organelle proximities are not a synthetic effect caused by the split Venus reporter, we used a second complementation system based on the engineered *Deinococcus radiodurans* infrared fluorescent protein IFP1.4 (from here after termed Far Red (FR)). This split FR fluorophore was previously shown to be reversible[20] and hence has much

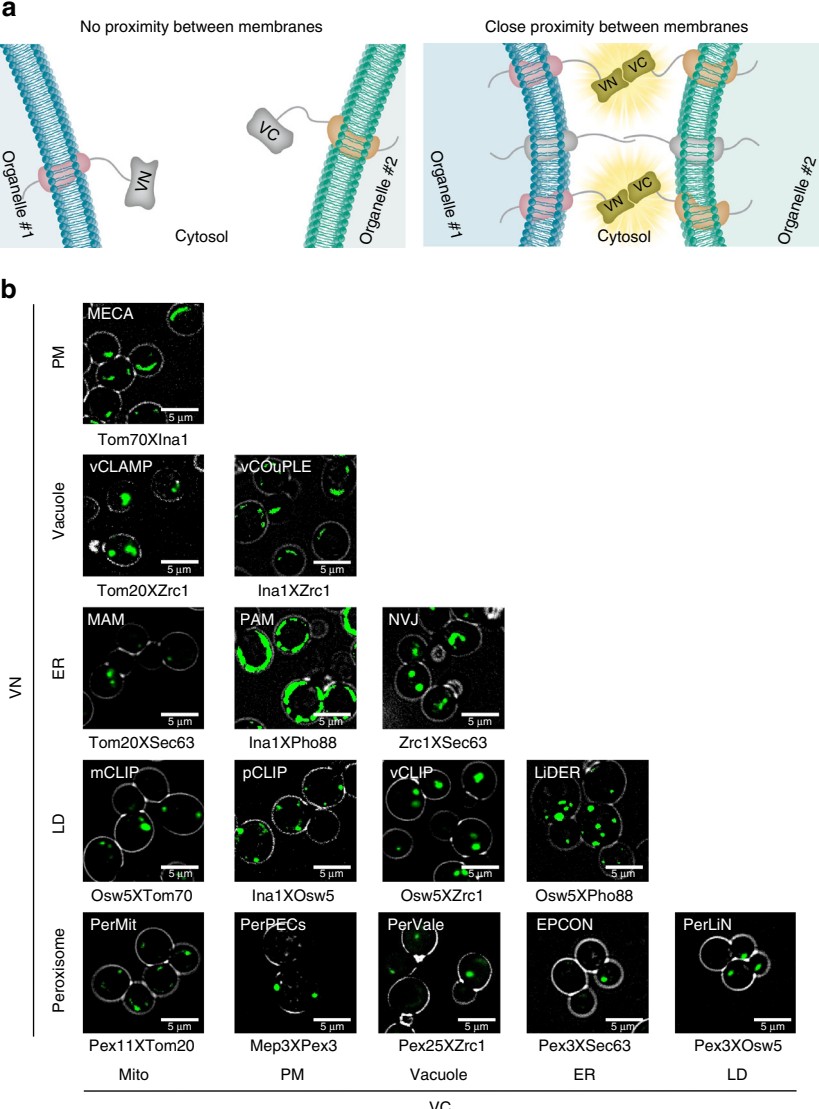

**Fig. 1** A Split Venus reporter uncovers four potential new contact sites. **a** A schematic diagram of the split Venus contact site reporter. We fused half of a Venus fluorescent protein (VN) to a membrane protein localized to one organelle and the other half of the Venus protein (VC) to a membrane protein localized to a second organelle. Only when the two organelles come into close proximity, such as that which occurs at a contact site, a fluorescent signal is emitted. **b** Pairwise combinations between membrane proteins tagged with half of the Venus protein were used to detect the proximity between cellular compartments (mitochondria—Mito, plasma membrane—PM, vacuole, endoplasmic reticulum—ER, lipid droplets—LD and peroxisomes). The names of previously identified contacts are written in white. The pattern and abundance of the different reporters is variable. This systematic analysis shows that every two cellular compartments that were examined can be juxtaposed suggesting four new contact sites. We suggest naming the new contacts (in bold white): vCOuPLE (vacuole-plasma membrane contact); pCLIP (for plasma membrane contact with lipid droplets); PerPECs (for peroxisome plasma membrane contact site), and PerVale (for peroxisome vacuole contact)

lower intrinsic affinity than the split Venus reporter. Indeed, the MAM FR reporter did not complement loss of ERMES like the split Venus reporter did (Supplementary Fig. 1j). Indeed, while we could use the split FR to visualize known contact sites such as the NVJ, MAM, ER-PM, and vCLAMP (Supplementary Fig. 3a), the low quantum yield of the FR fluorophore makes the split FR reporter much harder to visualize. Hence, we did not continue working with it for the below screens.

Finally, our systematic analysis uncovered four potential new contact sites, never before identified or described in any eukaryote (Fig. 1b). We therefore first validated them using the split FR reporter (Supplementary Fig. 3b). Following this verification we suggest the existence of a vacuole-PM contact that we now name vCOuPLE (vacuole-plasma membrane contact); A PM-LD contact site that we now name pCLIP (for plasma membrane contact with lipid droplets); A peroxisome-PM contact site that we now name PerPECs (for peroxisome plasma membrane contact site); and a peroxisome-vacuole contact site that we now name PerVale (for peroxisome vacuole contact) (Fig. 1b).

**Studying peroxisome-mitochondria contacts**. One of the organelles with the least studied contact sites is the peroxisome; a small, ubiquitous organelle that participates in central pathways of cellular metabolism[21]. Although several peroxisome contacts sites were previously identified in various organisms and were shown to have different functions including organelle inheritance and lipid transfer[17,22–25], it was clear that more contacts, tether complexes, and contact functions, await to be discovered.

One of the organelles with which peroxisomes have a tight interplay is mitochondria[22,26]. Peroxisomes and mitochondria share proteins responsible for their division machinery[27,28] and exhibit a tight metabolic cooperation in β-oxidation of fatty acids. Recently it was shown that mammalian peroxisome biogenesis can occur from mitochondria[29]. Loss of optimal β-oxidation capacity leads to severe metabolic disorders[30–33] suggesting that transfer of solutes through contact sites of peroxisomes and mitochondria would be of central metabolic importance. Despite the tight peroxisome–mitochondria relationship, the mechanisms of communication between the two organelles are still elusive, but diffusion processes, vesicular transport, and physical contact sites have all been implicated to be involved in this relationship[34–37].

We set to systematically characterize the Peroxisome-Mitochondria (PerMit) contact site by testing different pairs of the PerMit fluorescent reporters, enabling us to avoid reporter-specific effects (Fig. 2a, Supplementary Fig. 3c). We could observe that all reporters gave a similar signal as did a PerMit FR reporter (Supplementary Fig. 3d). Additionally, since the length of the cytosolic domains of the various reporter fusions we used ranged from 12.9 to 71.55 nm (assuming that a chain is forming an α-helix coil) and no difference in signal size or intensity could be observed, this verifies a distance range of 10–80 nm of the contact as has previously been suggested for other contacts.

We first verified that the PerMit reporters are localized to the interface between peroxisomes and mitochondria and do not impair the organelles, by co-expressing the reporters together with a peroxisomal marker and a mitochondrial marker (Fig. 2b, Supplementary Fig. 3e). Moreover, we could show that the PerMit reporter accurately marked the real contact sites by its co-localization with a specific sub-domain in the mitochondrial matrix that is enriched with the pyruvate dehydrogenase (PDH) complex and is also localized in proximity to the ER-mitochondria contact site on the mitochondrial outer membrane as we have previously characterized for this contact[38] (Fig. 2c, d). To exclude the possibility that the PDH and ERMES proteins are artificially re-targeted to the PerMit reporter site, we induced the expression of the reporter (that had one half on a galactose inducible promoter) and microscopically followed the PDH and ERMES localization. We could then clearly see that the reporter signal accumulated next to existing sites of PDH and ERMES and not vice versa (Fig. 2e). This result proves that the reporter localizes correctly to existing contact sites in their physiological local.

However, it was clear that our Venus reporter stabilized existing contacts since we found an increase of co-localization between mitochondria and peroxisomes when the reporter was expressed (Supplementary Fig. 3f). Hence while the Venus reporter does not create random artificial contacts it does stabilize them and this was taken into consideration in our future analysis.

**Identifying tethers and regulators of PerMit.** It was previously suggested that in S. cerevisiae Pex11, a key protein involved in peroxisome proliferation, and Mdm34, part of the ERMES complex that tethers the ER-mitochondria contact, serve as a peroxisome-mitochondria tether complex[39]. Additionally, it was suggested that in mammals the ATP binding cassette (ABC) transporter 1 (ABCD1), located to the peroxisomal membrane, and whose loss of function causes X-linked adrenoleukodystrophy (X-ALD), is a peroxisome-mitochondria tether[40]. Recently, it was suggested that acyl-coenzyme A-binding domain (ACBD2/ECI) isoform A mediates peroxisome-mitochondria interaction in mouse tumor Leydig cells[37]. Despite the fact that several PerMit tethering molecules have already been suggested, all contact sites studied were shown to have several tether complexes each with a unique function and regulation. Hence, we wondered if additional tethers and regulators of the PerMit contact could be found in yeast.

Since, as shown above, single gene deletions were often not enough to cause disassembly of the contact site reporter signal, we searched for potential tethering molecules by screening for proteins that expanded the extent of the reporter signal when overexpressed (similarly to the expansion visualized upon over-expression of Vam6 or Lam6 (Supplementary Fig. 1h, i)). We took into consideration that overexpression of one tethering protein would lead to contact expansion only if the amount of its partner protein or lipid on the opposing membrane is not a limiting factor, or if it is mediating contact through a homotypic interaction. Hence for this reason, and others, our screen may not have been saturating.

Using an automated mating procedure we created a collection of ~1800 strains each expressing one mCherry-tagged yeast protein that is expressed under a strong (TEF2) promoter[41] (including all peroxisomal and mitochondrial outer membrane proteins) and integrated the PerMit reporter to their genome. We then screened the cells by high content microscopy and found 43 strains that caused expansion of the PerMit reporter signal (Fig. 3a, Supplementary Table 2). As we were looking for tethers and direct regulators we concentrated on proteins that are localized to either peroxisomes or mitochondria and by that we narrowed the list to 12 proteins (Fig. 3b). Following several secondary screens, we decided to further focus on two proteins and study their potential tethering capabilities. The first, Pex34, is a peroxisomal membrane protein conserved to humans with as yet unidentified molecular function. The second is the yeast mitofusin, Fzo1, an outer mitochondrial membrane protein that functions in mitochondrial tethering and fusion and whose mammalian homologue, Mitofusin 2 tethers the ER and mitochondria[42].

Notably, when PEX34 or the previously suggested PerMit tether, PEX11, were deleted, we could not detect a clear reduction of the PerMit reporter signal (Supplementary Fig. 4a, b). Although Fzo1 deletion did reduce the PerMit reporter signal it also dramatically altered the shape of mitochondria and hence the effect observed by the PerMit reporter could be indirect. The lack of effect of PEX34 or PEX11 deletion emphasizes the strength of using an overexpression strategy and not a single deletion strategy.

Importantly, both Fzo1 and Pex34 overexpression also enhanced the split FR reporter (Supplementary Fig. 4c). Moreover, their overexpression was sufficient to tether peroxisomes and mitochondria in the absence of any reporter (Figs. 3c, d and 4b). In addition, by following the organelle movement we found that overexpressing Fzo1 or Pex34, even in the absence of the PerMit reporter, reduced the movement speed of peroxisomes in the cell similarly to expressing the PerMit reporter alone (Fig. 3e). Since tethering reduces motility, this supports their role in organelle tethering. Hence Fzo1 and Pex34 both have the characteristics of potential PerMit tethers leading us to further characterize them. While the gold standard of contact sites is to visualize them by electron microscopy (EM), growth of the cells in presence of glucose render peroxisomes difficult to detect by EM. In fact, we had to screen several thousands of cell sections to be able at the end to only see a handful of distinctive peroxisomes. Hence, this system can not lend itself to a statistically-relevant quantitation for the relevance of our potential tethers on contact site formation.

**Fzo1 is a PerMit tether protein.** Fzo1 is the yeast mitofusin protein and a homolog of the human mitofusins 1 and 2 (MFN1

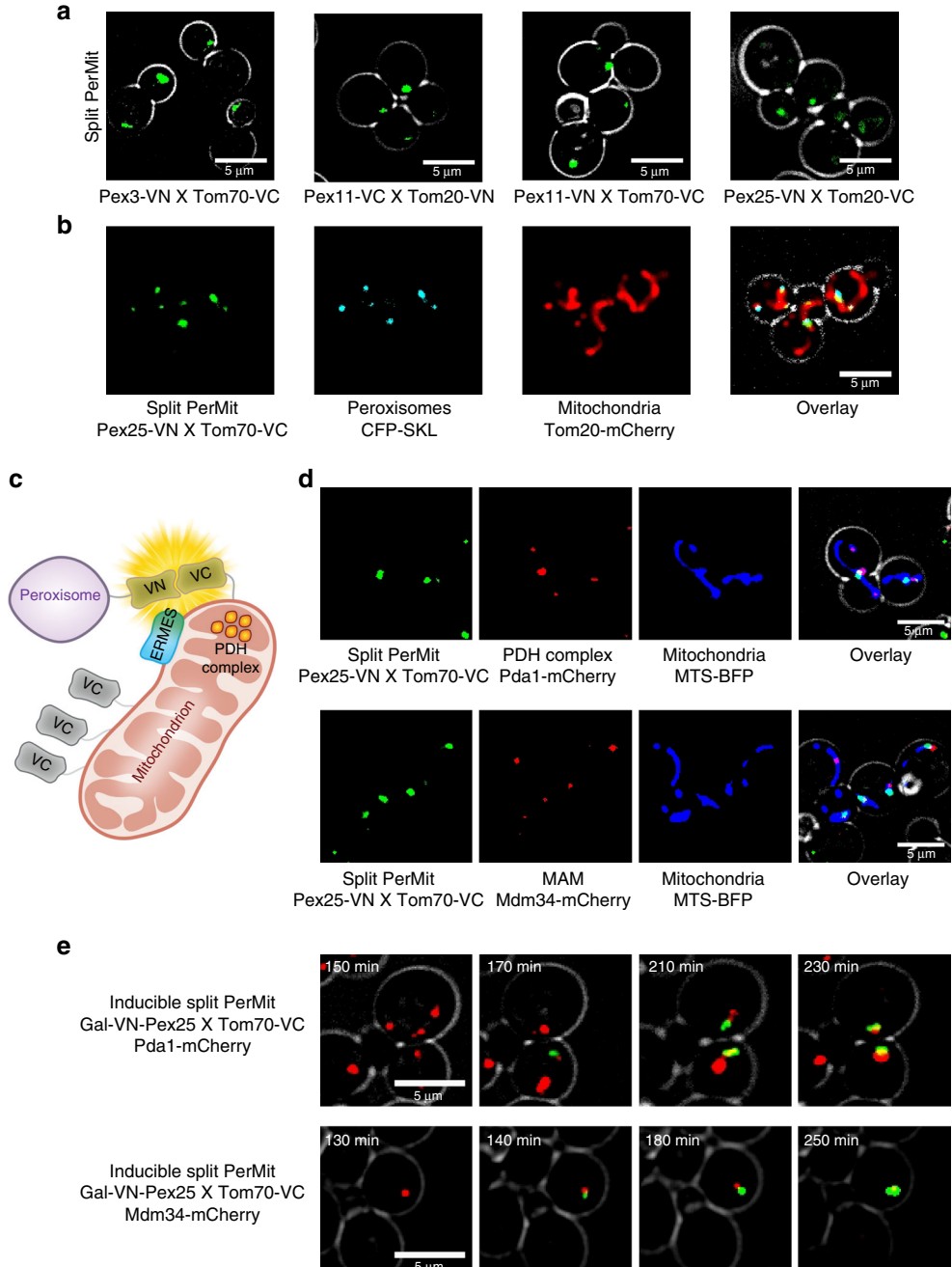

**Fig. 2** The Peroxisome-Mitochondria (PerMit) reporter accurately reports on the contact site between the organelles. **a** Pairwise combinations of three different peroxisomal membrane proteins (Pex3/Pex11/Pex25) tagged with one half of the split Venus protein and one of two mitochondrial membrane proteins (Tom70/Tom20) tagged with the complementary split Venus half, all lead to a fluorescent signal indicating that the two organelles are in close proximity and that the signal is not dependent on the marker protein used. **b** The PerMit reporter signal co-localizes to both a peroxisomal marker (CFP-SKL) and to a mitochondrial marker (Tom20-mCherry) demonstrating that it indeed marks sites of close apposition between the two membranes. **c** A schematic diagram showing that peroxisomes are found in proximity to specific niches in mitochondria: the ER-Mitochondria contact site (represented by the ERMES tether complex) and to a mitochondrial niche in which the pyruvate dehydrogenase complex (PDH complex) is enriched. **d** The PerMit reporter indeed localizes in proximity to the mitochondrial (mitochondria are marked by MTS-BFP) niche that is enriched with PDH complexes (marked by Pda1-mCherry) and to ER-mitochondria contact sites (marked by Mdm34-mCherry) suggesting that it reports on real contact sites and not random sites of proximity. **e** Time point images (the time is indicated in white) of PDH complexes (marked by Pda1-mCherry) or ER-mitochondria contact sites (marked by Mdm34-mCherry) after shifting the cells to galactose (time:0) thus inducing the PerMit reporter by expressing its peroxisomal half (Gal-VN-Pex25). The images show that the PerMit reporter is formed next to existing Pda1 or Mdm34 niches

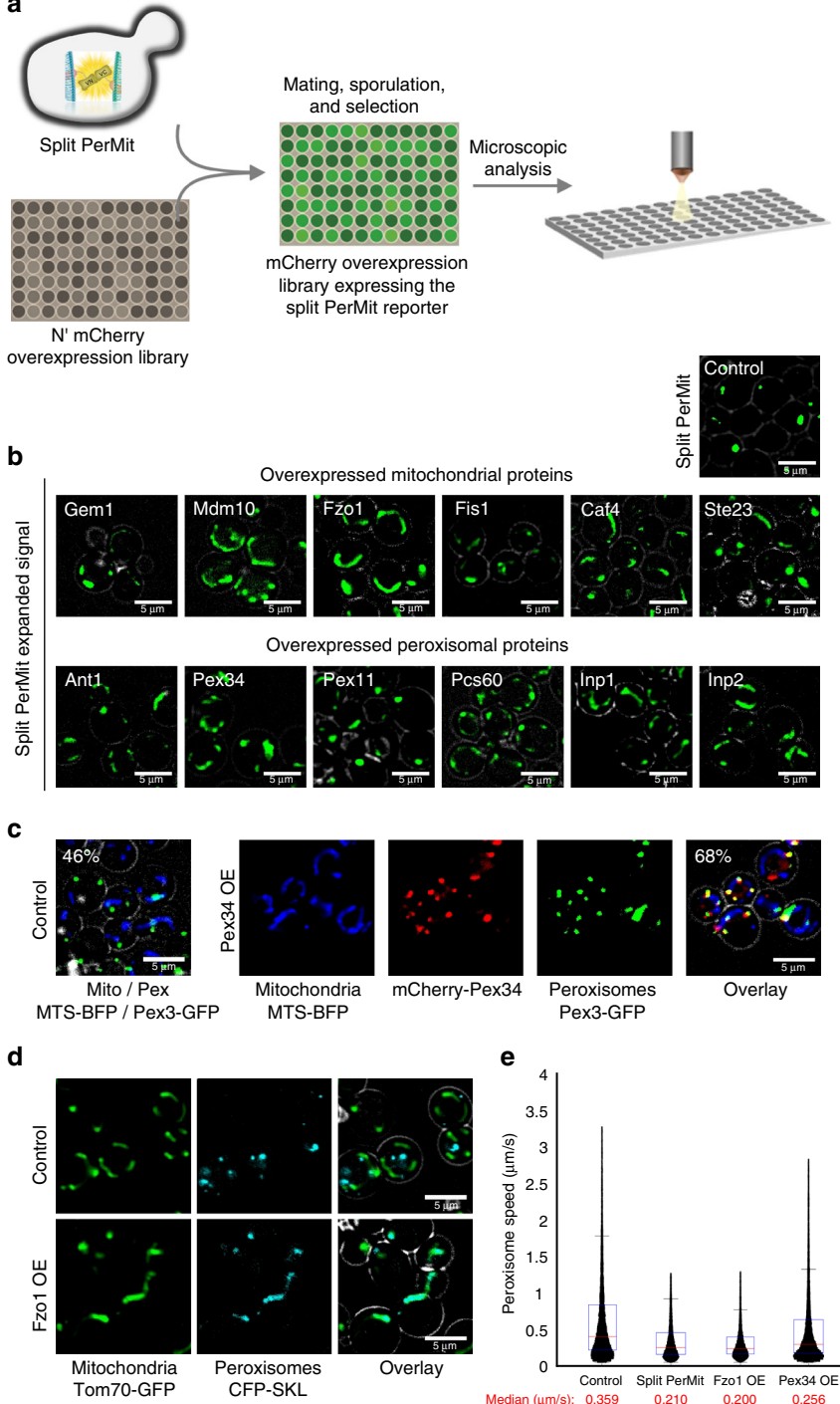

**Fig. 3** A high content microscopy screen reveals proteins that expand the PerMit reporter signal when overexpressed. **a** A schematic representation of the microscopy screen. Yeast strains carrying the PerMit reporter were mated with a collection of strains each expressing one protein tagged with a mCherry fluorophore while being expressed under a strong, *TEF2*, promoter. Haploid cells carrying the reporter and an overexpressed protein tagged with mCherry were analyzed using a fluorescent microscope aiming to find strains in which the reporter signal was expanded. **b** Representative images of either mitochondrial or peroxisomal genes whose overexpression led to the expansion of the PerMit reporter signal. **c** Pex34 overexpression (Pex34 OE) leads to increased co-localization of peroxisomes (marked by Pex3-GFP) to mitochondria (marked by MTS-BFP) independently of the presence of the PerMit reporter. White numbers—percentage of peroxisomes that co-localize with mitochondria. **d** Fzo1 overexpression (Fzo1 OE) leads to peroxisomes (marked by CFP-SKL) clumping onto mitochondria (marked by Tom70-GFP) in the absence of the PerMit reporter. **e** Mean peroxisome speed was calculated from 4 min time-lapse videos of strains carrying marked peroxisomes (CFP-SKL) without (control) or with the PerMit reporter and in Fzo1 OE or Pex34 OE cells without the PerMit reporter. Boxplots are shown with 25,000 point distributions from three independent experiments. Red lines indicate the median (the median values are written beneath each strain name). Bottom and top edges of the blue boxes indicate the 25 and 75 percentiles. The whiskers extend to the most extreme data points, not considering the outliers. The analysis shows that the PerMit reporter itself, as well as overexpression of Fzo1 or Pex34, reduce peroxisome motility

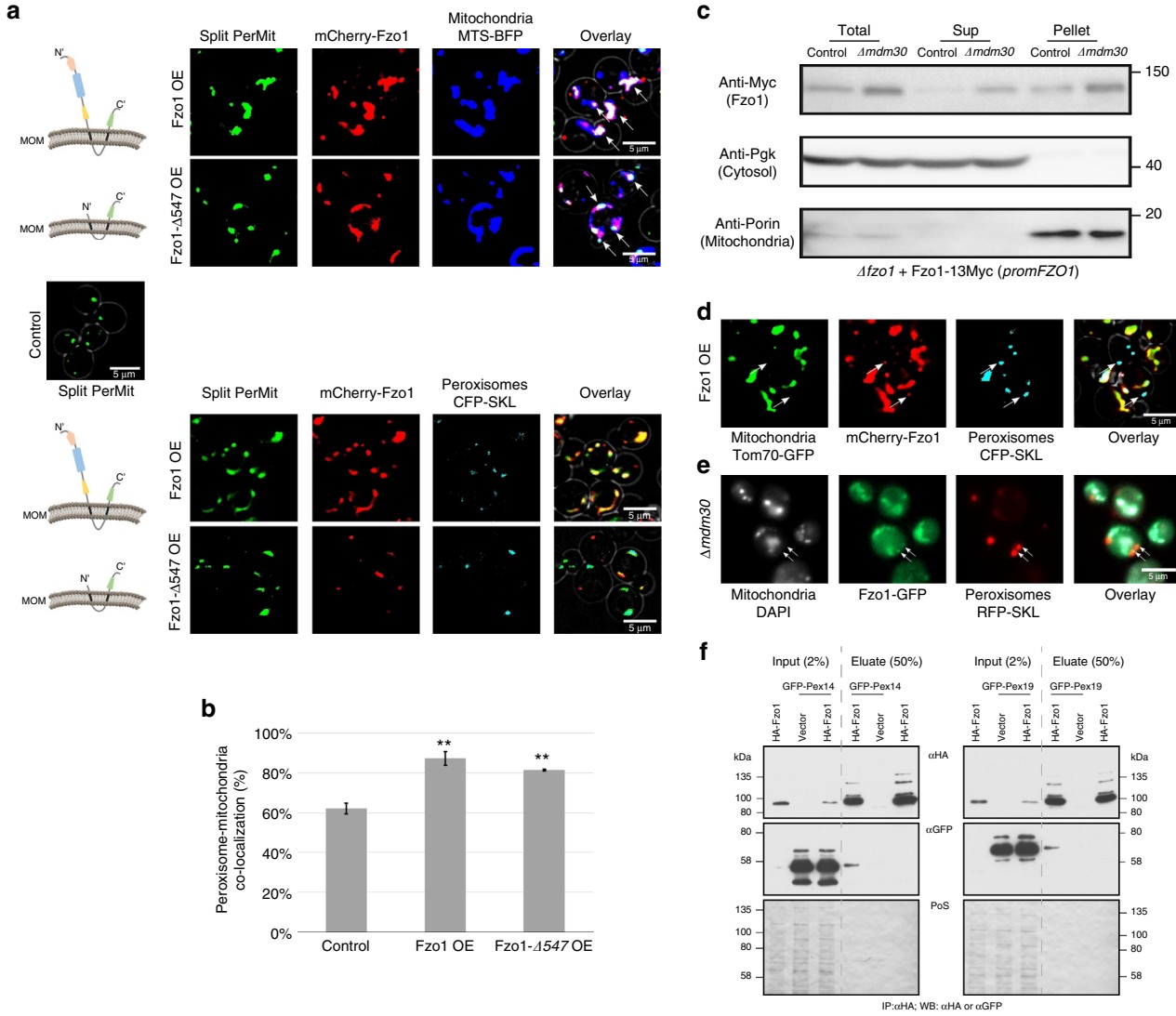

**Fig. 4** Fzo1 has characteristics of a tether for the PerMit contact site. **a** Overexpression of full-length (mCherry-Fzo1) or truncated (mCherry-Fzo1 Δ547) Fzo1 (control cells and schemes of Fzo1 on left). When full-length Fzo1 is overexpressed, mitochondria (marked by MTS-BFP) are hyper-fused (as previously described) yet Fzo1 is still only localized to mitochondrial patches in proximity to peroxisomes and to the contact (white arrows). All PerMits co-localize with Fzo1. Overexpression of truncated Fzo1 does not alter mitochondrial morphology, is sufficient for mediating PerMit expansion and enables Fzo1 localization to the PerMit site. Overexpression of full length or truncated Fzo1 does not affect peroxisomes (marked by CFP-SKL) number and size. **b** Quantification of peroxisomes that co-localize with mitochondria in control, full length or truncated Fzo1 overexpression. Bars represent the mean ± s.e. from three independent experiments. one-tail Student's $t$-test, $**p < 0.01$. **c** Total lysates from control and Δ$mdm30$ cells, in which native Fzo1 is tagged with 13Myc epitopes (Fzo1-13Myc), were subjected to subcellular fractionation yielding soluble (Sup) and mitochondria-enriched (Pellet) fractions. The distinct fractions were analyzed by immunoblotting with anti-Myc (Fzo1), anti-Pgk (a cytosolic marker) and anti-Porin (a mitochondrial marker) antibodies. Fzo1 is found in a non-mitochondrial fraction more readily visible when a slight increase in Fzo1 levels is created by deleting *MDM30*. **d** Fzo1 overexpression by a strong *TEF2* promoter demonstrates Fzo1 signals that do not co-localize with mitochondria (marked by Tom70-GFP) but rather with peroxisomes (marked by CFP-SKL) (white arrows). **e** *MDM30* depletion demonstrating that Fzo1-GFP is localizes to both mitochondria (marked by DAPI) and peroxisomes (marked by RFP-SKL) (white arrows). **f** A physical interaction between Fzo1 and Pex14/Pex19. HA-Fzo1 or its corresponding empty vector were expressed in control, GFP-Pex14, or GFP-Pex19 strains. HA-Fzo1 was immunoprecipitated from solubilized crude membrane extracts using HA-coupled beads. Eluted Fzo1 was analyzed by SDS-PAGE and immunoblotting using anti HA- or GFP-specific antibodies. Ponceau S staining (PoS) was used to compare protein levels

and MFN2) whose domain architecture has been well studied[43]. It was previously shown that mitofusins mediate mitochondria–mitochondria tethering (as well as fusion)[44–46]. In addition, a specific role of MFN2 in tethering mitochondria to other organelles including the ER[47], melanosomes[48] and LDs[49] was demonstrated. Hence we hypothesized that Fzo1 could also mediate peroxisome-mitochondria tethering.

We first verified that the effect of Fzo1 overexpression on the PerMit expansion was independent of the PerMit reporter combination used (Supplementary Fig. 5a). Since Fzo1 fuses mitochondria its overexpression leads to hyper-fused mitochondrial networks[45,50] (Supplementary Fig. 5b, c) that may affect all mitochondria contact sites in a non-specific manner. Arguing against this was the observation that the effect of Fzo1

overexpression was specific to the PerMit contact (Supplementary Fig. 5d). Additionally, the overexpression of Fzo1 did not affect other peroxisomal contacts (Supplementary Fig. 5e) nor Pex25, the peroxisomal protein that was used as part of the PerMit reporter (Supplementary Fig. 5b). However, the specific effect of Fzo1 on the PerMit still is not enough to determine that it is direct. Hence, we set out to ascertain this.

We have recently suggested three criteria to define a protein as a true, direct, tethering molecule at a contact site[2]. Exploring Fzo1 we found that it abides by all three criteria:

The first criterion is structural capacity: To have a tethering capacity, proteins must have domains that enable binding to both organelles. The domain that is utilized for mitochondrial tethering in Fzo1 is the Heptad Repeat 2 (HR2) domain[51]. We therefore assayed whether this domain is also important for the PerMit tether. Indeed, overexpressing of a truncated form that had only the HR2 domain but was lacking the Heptad Repeat at the N′ (HRN), GTPase, and HR1 domains (Δ547), was sufficient, to expand the PerMit signal (Fig. 4a). Moreover, it enhanced the recruitment of peroxisomes to mitochondria even in the absence of the reporter with similar extent to the full-length Fzo1 (Fig. 4b). Importantly, expressing the mutant Fzo1 did not alter mitochondria shape (Supplementary Fig. 5c) excluding the possibility that the PerMit expansion is a result of altering mitochondria shape.

The second criterion is enrichment at the contact site: To be a real tether, a protein must be enriched in, or localized exclusively to the contact site. Indeed, both full-length Fzo1 as well as the truncated version were not homogenously distributed on mitochondria (Fig. 4a, Fzo1 does not fully co-localize to the mitochondrial marker) but rather were enriched at PerMit patches indicating accumulation in areas of mitochondria-peroxisome proximity (Fig. 4a, white arrows). Moreover, all PerMit signals co-localized with Fzo1 (Fig. 4a) implying that Fzo1 has a defined localization to PerMit contacts and that no PerMit contact occurs without Fzo1 being expressed at the same place.

The third criterion is functional activity: To be a bona fide tether, a protein must bring together the two opposing membranes. As shown above overexpression of Fzo1 brought into close proximity the two organelles even in the absence of the reporter (Fig. 3d). It was previously suggested that mitofusin tethering is mediated by homotypic interaction between molecules that are localized to the opposing membranes[46,48,51–53]. Hence, we wondered if this could also be the mechanism by which Fzo1 is mediating peroxisome-mitochondria tethering. Until now, a non-mitochondrial fraction of Fzo1 has never been reported. However, many mitochondrial outer membrane proteins, such as Fis1 and Msp1, are dually localized to both mitochondria and peroxisomes[54–57]. Hence, we performed careful fractionation assays that demonstrated that indeed there is a small, but significant and reproducible, fraction of Fzo1 that is found outside of mitochondria (Fig. 4c). Moreover this fraction became more readily visible when a slight increase in Fzo1 levels was created by deleting its regulator Mdm30[44,50,58,59]. To find whether this non-mitochondrial fraction co-localized with peroxisomes we visualized Fzo1 either expressed under a strong (TEF2) promoter, or on the background of Δmdm30, and could observe an Fzo1 population that did not co-localize with mitochondria but rather co-localized with peroxisomes in both strains (Fig. 4d, e). In support of Fzo1 localizing to the peroxisome membrane, we found the peroxisomal membrane protein targeting machinery (Pex19) as well as the membrane insertase of matrix proteins (Pex14) as binding partners of Fzo1 (Fig. 4f).

Since Fzo1 abided by all three criteria, we suggest that it functions as a bona fide tether of the PerMit contact site. The tethering function of Fzo1 might be mediated by a homotypic interaction between mitochondrial Fzo1 and peroxisomal Fzo1.

However, since we could not detect Fzo1 on peroxisomes when expressed at endogenous levels, we cannot exclude the possibility that the interaction between the organelles is mediated by binding of mitochondrial Fzo1 to another peroxisomal protein.

**Pex34 tethering affects β-oxidation**. Pex34 is a 144 amino-acid peroxisomal membrane protein that affects peroxisome growth and division by an unknown mechanism[60]. Since Pex34 shares a remote homology with Pex11 (using HHpred[61]), that was previously suggested to serve as a PerMit tether[39], we examined if Pex34 also has the capacity to directly tether the PerMit contact.

To this end, we found that Pex34 overexpression leads to expansion of the PerMit signal independently of the reporter combination (Supplementary Fig. 6a). In addition, we verified that the effect of Pex34 overexpression on the PerMit reporter is not simply a result of affecting the expression of the two halves of the reporter (Supplementary Fig. 6b). Moreover Pex34 overexpression had no effect on mitochondrial shape (Supplementary Fig. 6c, d), on the ability to respire on glycerol containing media (Supplementary Fig. 6e), nor on the MAM contact site (Supplementary Fig. 6f). However, peroxisomes were more numerous in agreement with previous observations[60] (Supplementary Fig. 6g). Importantly, the increased number of peroxisomes was not the reason for more PerMit contacts as no other peroxisomal contact site reporter was affected (Supplementary Fig. 6h).

Importantly, Pex34 was also not found on the entire peroxisomal surface but rather was enriched in niches that co-localized with the PerMit signal implying that Pex34 has a defined localization in the PerMit contact (Supplementary Fig. 6g). Additionally, we excluded the possibility that mistargeting of overexpressed Pex34 is the cause for the expanded PerMit signal. This was true either when peroxisomes were intact or upon PEX3 deletion, when no mature peroxisomes exist (Supplementary Fig. 6i).

A true tether should affect the function of the contact site. In S. cerevisiae, peroxisomes are the sole organelles in which β-oxidation of fatty acids occurs. The final product of fatty acid degradation in peroxisomes, acetyl-CoA, subsequently has to be transported to mitochondria for complete degradation into $CO_2$ and $H_2O$ by the Krebs cycle. We previously observed that the PerMit contact is localized in vicinity to a niche in the mitochondrial matrix in which the PDH complex is enriched[38]. Therefore, we suspected that one function of the PerMit contact is to enable efficient transport of acetyl-CoA molecules from peroxisomes to mitochondria where they can be utilized for respiration.

In support of the PerMit contact being important for the transfer of β-oxidation products, we found that peroxisome proximity to mitochondria increased when yeast were grown on oleate as a sole carbon source (Fig. 5a). The increase in the PerMit extent was not simply a result of an increase in peroxisome number since in ethanol the number of peroxisomes was similar to oleate however, the number of PerMit foci was only mildly increased (Fig. 5b).

To further examine the possible involvement of the PerMit contact in transfer of β-oxidation products, we biochemically measured the rate of acetyl-CoA transfer to mitochondria and its conversion to $CO_2$ by measuring acid soluble products (ASPs) and $CO_2$ production after incubating the yeast cells with radiolabeled [1-C14] octanoate (C8:0). First, we verified that the PerMit reporter itself does not affect β-oxidation (Fig. 5c). We then measured the effect of overexpressing the two tether proteins: Fzo1 and Pex34, in the reporter strains. We found that overexpression of Fzo1 did not result in an increase in $CO_2$

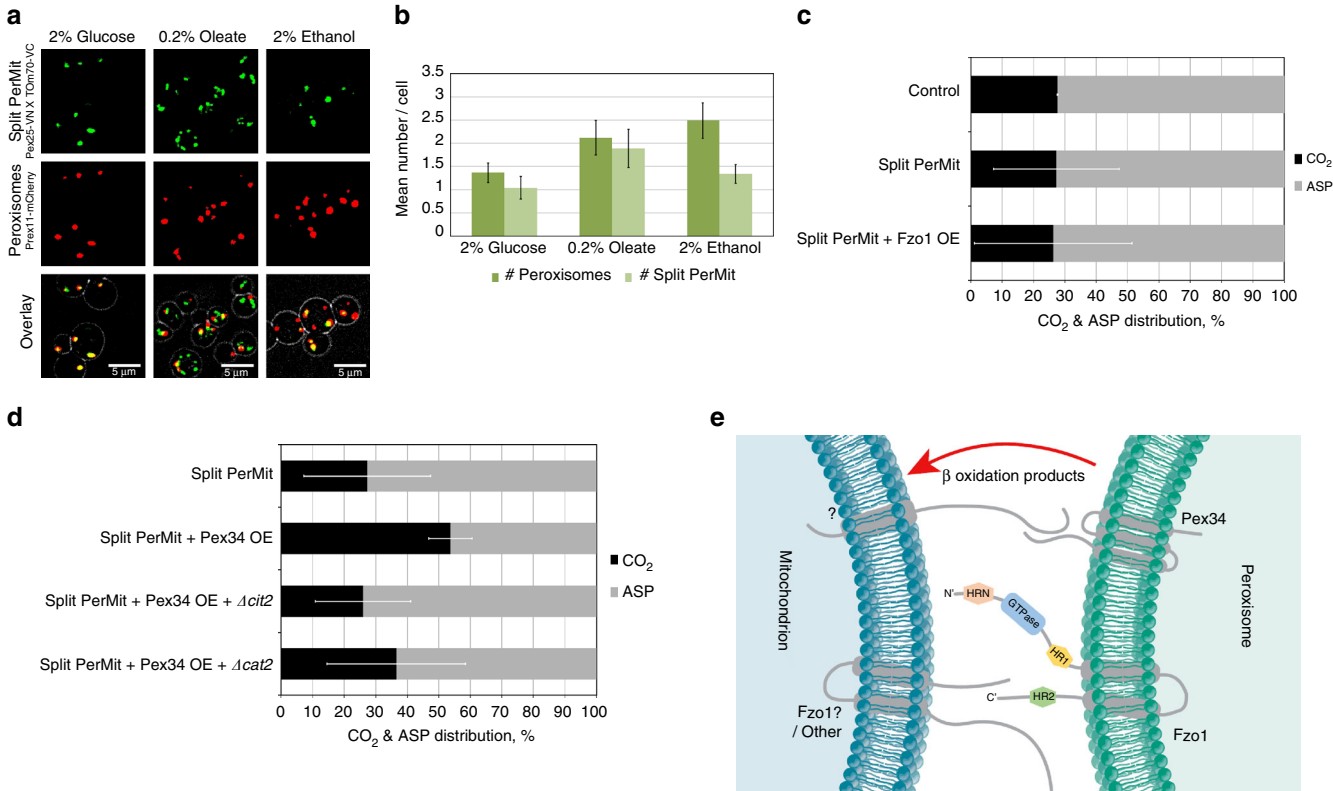

**Fig. 5** The PerMit contact site contributes to β-oxidation. **a** Cells expressing the PerMit reporter and a peroxisomal marker (Pex11-mCherry) were grown in media containing either glucose, oleate, or ethanol as a sole carbon source. Representative images in different media are shown. **b** Quantification of the number of Pex11-mCherry and PerMit reporter puncta per cell showing that the number of PerMit contacts relative to the number of peroxisomes is increased in oleate. Data are presented as mean ± s.d. from four independent experiments, 100 cells per experiment. **c**, **d** Strains were grown on oleate and were assayed for β-oxidation activity. The $CO_2$ production and the acid soluble products (ASPs) were used to quantify fatty acid oxidation. Results are presented as percentage relative activity to the rate of oxidation in a control strain. Data are presented as mean ± s.d. from three or more independent experiments. **e** A schematic hypothesis model of the new findings regarding the PerMit contact site. Pex34 should interact with an unknown mitochondrial protein (marked by "?") to mediate the contact site and to potentially enable metabolite transfer of β-oxidation products. Fzo1 on mitochondria may mediate tethering either by interacting with the peroxisomal pool of Fzo1 or with another peroxisomal protein

production (Fig. 5c). In contrast, overexpression of Pex34 resulted in a marked increase in $CO_2$ production (Fig. 5d), indicating that PerMit expansion by Pex34, but not by Fzo1 or by the reporter, stimulates the transport of acetyl-CoA from peroxisomes to mitochondria.

Two pathways for transport of acetyl-CoA to mitochondria exist[62]. The first pathway involves peroxisomal conversion of acetyl-CoA into citrate by peroxisomal citrate synthase (Cit2), which can then be transported to mitochondria. The second pathway involves peroxisomal conversion of acetyl-CoA into acetylcarnitine by carnitine transferase (Cat2), which is then transported to mitochondria. To study which of the two acetyl-CoA export pathways relies on the Pex34-mediated PerMit expansion, we overexpressed Pex34 with the PerMit reporter in $\Delta cit2$ or $\Delta cat2$ cells which each abolish one route of transport. The increased $CO_2$ production in the Pex34 overexpressing cells was abolished by deleting the citrate synthase ($\Delta cit2$) and reduced when deleting the acetylcarnitine transferase ($\Delta cat2$) (Fig. 5d). This suggests that Citrate is the prominent molecule transferred through the Pex34 expanded contacts. The observation that increased tethering itself by either the reporter or Fzo1 over-expression did not affect carbon transfer, whereas overexpression of Pex34 did, together with the observation that inhibiting acetyl-CoA conversion into Citrate abolishes the effect of Pex34 overexpression, lead us to suggest that Pex34 is a more specific

tether functioning in transfer of β-oxidation intermediates between compartments.

The observation that Fzo1 overexpression did not affect $CO_2$ production implies that different tether complexes that hold peroxisomes and mitochondria have different functions. Supporting the existence of different tether complexes, we found that deletion of *FZO1* did not affect the ability of Pex34 to expand the PerMit contact and vice versa (Supplementary Fig. 7a, b). Hence we suggest that Pex34 and Fzo1 are not parts of the same tether complex. While Pex34 is a potential tether involved in β-oxidation, the functional significance of the Fzo1 tether awaits discovery.

## Discussion

Organelles, once studied as isolated structures specialized each in their own specific functions, are now appreciated for their tight inter-connectivity and cross-talk with all other cellular parts. Within the cellular milieu, organelles must work in concert and intensively communicate with each other to enable life. In the last years it has become apparent that one way of communication between organelles is by physical contact sites and a current goal of cell biology is to identify the extent and functional significance of contact sites.

Here we used a split Venus complementation assay, to identify and characterize contact sites with no need for any prior

knowledge on the presence of the contact site, its tethers, or resident proteins. Using this tool, we identified four potential novel contact sites in yeast. Future studies using EM, functional studies and biochemical reconstitution will be required to verify the existence of those new contacts and to characterize them. However, this strongly suggests that every organelle, at some point, creates contact sites with any other organelle in the yeast cell. Together with recent studies in mammalian cells[17] those observations dramatically change the historical view of contact sites as rare occurrences and as structures mainly formed with the ER.

A unique aspect of the split fluorescence reporter is that it is universal and can be utilized in any cell type, organism or system. Hence our newly discovered yeast contact sites can easily be verified in mammalian cells.

Moreover, we performed a high content experiment that enabled us to identify potential tethers and regulators of the PerMit contact and further concentrated on two proteins: Pex34 and Fzo1. We suggest that both proteins are tether proteins as they fit the three "gold standard" criteria that we have previously defined for a tether protein[2]: they have a defined localization, a capacity to tether membranes and a functional activity. We also suggest that the physical interaction between peroxisomes and mitochondria contributes to β-oxidation of fatty acids and suggest that Pex34 tethering is specifically involved in this function (Fig. 5e).

It is known that contact sites are commonly held by several tether complexes and have different functions[3]. Hence it would be interesting to identify additional PerMit tethers as well as to uncover the mitochondrial binding partner of Pex34. Future work will be required to understand how Fzo1 is targeted to peroxisomes, the extent of interplay between Fzo1 and Pex34, which additional functions are carried out by the PerMit and by which molecules, and importantly how this contact is regulated. As these questions start to be tackled, it is clear that a contact that affects β-oxidation should have dramatic effects on human development and in disease[63].

## Methods

**Yeast strains and strain construction**. Yeast strains are all based on the BY4741 laboratory strain[64], except of Δmdm30 strains and their controls that are derivatives of W303 and the Δtether strain and control[15] that are derivatives of SEY6210.1. Genetic manipulations were performed using the lithium acetate, polyethylene glycol (PEG), single-stranded DNA (ssDNA) method for transforming yeast strains[65], using integration plasmids that are listed in Supplementary Table 3. All strains used in this study are listed in Supplementary Table 4.

**Primer design**. Primers for genetic manipulations and their validation were designed using Primers-4-Yeast (http://wws.weizmann.ac.il/Primers-4-Yeast)[66]. Primers for genetic tagging of split Venus proteins were designed using Primers-4-Yeast with flanking sequences Primers: F-GGTCGACGGATCCCCGGGTT R-TCGAT-GAATTCGAGCTCGTT[8]. Primers for genetic tagging of split IFA1.4 (split FR) proteins were designed using Primers-4-Yeast with flanking sequences Primers: F-GGCGGTGGCGGATCAGGAGGC R- TTCGACACTGGATGGCGGCGTTAG[20]

**High content screening**. To create collections of haploid strains containing both the PerMit reporter (Pex25-VN; Tom70-VC) and an overexpressed, mCherry tagged, protein a query strain was constructed on the basis of a Synthetic Genetic Array (SGA) compatible strain. Using automated mating approaches[67,68], the PerMit reporter query strain was crossed with a SWAT N'-mCherry library[41] which is a collection of ~1800 strains. In each strain, one protein is N' tagged with mCherry and expressed under an overexpression promoter (TEF2pr). Yeast manipulations in high-density format were performed on a RoToR bench top colony arrayer (Singer Instruments). In short: mating was performed on rich medium plates, selection for diploid cells was performed on SD$_{MSG}$-His plates containing Geneticin (200 μg ml$^{-1}$) and Nourseothricin (200 μg ml$^{-1}$). Sporulation was induced by transferring cells to nitrogen starvation media plates for 7 days. Haploid cells containing the desired mutations were selected by transferring cells to SD$_{MSG}$-His plates containing Geneticin (200 μg ml$^{-1}$) and Neurseothricin (200 μg ml$^{-1}$), alongside the toxic amino-acid derivatives Canavanine and Thialysine

(Sigma-Aldrich) to select against remaining diploids, and lacking Leucine to select for spores with an alpha mating type.

The collections were visualized using an automated microscopy setup[69]. In brief, cells were transferred from agar plates into 384-well polystyrene plates for growth in liquid media using the RoToR arrayer. Liquid cultures were grown in a LiCONiC incubator overnight at 30 °C in SD$_{MSG}$-His plates containing Geneticin (200 μg ml$^{-1}$) and Nourseothricin (200 μg ml$^{-1}$). A JANUS liquid handler (PerkinElmer) connected to the incubator was used to dilute the strains to an OD$_{600}$ of ~0.2 into plates containing SD medium (6.7 g l$^{-1}$ yeast nitrogen base and 2% glucose) supplemented with complete amino acids. Plates were incubated at 30 °C for 4 h in SD medium. The cultures in the plates were then transferred by the liquid handler into glass-bottom 384-well microscope plates (Matrical Bioscience) coated with Concanavalin A (Sigma-Aldrich). After 20 min, wells were washed twice with SD-Riboflavin complete medium to remove non-adherent cells and to obtain a cell monolayer. The plates were then transferred to a ScanR automated inverted fluorescent microscope system (Olympus) using a robotic swap arm (Hamilton). Images of cells in the 384-well plates were recorded in SD-Riboflavin at 24 °C using a ×60 air lens (NA 0.9) and with an ORCA-ER charge-coupled device camera (Hamamatsu). Images were acquired in two channels: GFP (excitation filter 490/20 nm, emission filter 535/50 nm) and mCherry (excitation filter 572/35 nm, emission filter 632/60 nm). After acquisition, images were manually reviewed using the ScanR analysis program. Hits form the library were imaged a second time using spinning disk microscopy (as below).

**Manual microscopy**. Manual microscopy was performed using VisiScope Confocal Cell Explorer system, composed of a Zeiss Yokogawa spinning disk scanning unit (CSU-W1) coupled with an inverted Olympus IX83 microscope. Images were acquired using a ×60 oil lens and captured by a connected PCO-Edge sCMOS camera, controlled by VisView software, with wavelength of 488 nm (GFP/Venus), 561 nm (mCherry), 405 nm (BFP/CFP), and 640 nm (Far-red/IFP1.4). Images were transferred to ImageJ (http://imagej.net/Fiji/Downloads), for slight, linear, adjustments to contrast and brightness. Brightfield channel was used to segment the cells for image visualization of the yeast cells. The brightness was reduced to visualize only the exterior halo of the cell. Notably this halo has a larger circumference than the plasma membrane.

For Fig. 4e, Δmdm30 strain in which the chromosomal copy of FZO1 was tagged with GFP (FZO1-GFP cells) were transformed with an RFP-SKL expression plasmid (pRS316-mRFP-SKL). Resulting strains were grown to early exponential growth phase in SD-URA at 30 °C and mitochondrial DNA was labeled by incubating cells with 1 μg ml$^{-1}$ DAPI for 1 h. Cells were fixed with 4% formaldehyde and washed twice in water. Fluorescence microscopy was carried out with a Zeiss Axio Observer.Z1 microscope (Carl Zeiss S.A.S.) with a ×63 oil immersion objective equipped with the following filter sets: DAPI, GFP, and mCherry. Cell contours were visualized with Nomarski optics. Images were acquired with a SCMOS ORCA FLASH 4.0 charge-coupled device camera (Hamamatsu). Images were treated and analyzed with ImageJ.

**Split Venus reporters**. Proteins from each organelle were selected according to three criteria: (1) Being membrane spanning or anchored to ensure that they coat the outer surface of the organelle. Localization information was obtained from the literature as well as from the C' GFP library[70]. (2) Having the tagged termini facing the cytosol. The orientation was obtained from the literature as well from Phobius transmembrane topology tool (http://phobius.sbc.su.se/)[71,72]. (3) Expressed at high abundance. Protein abundance (molecules per cell) information was obtained from ref. [73]. We chose for tagging 26 candidate proteins from the following cellular compartments: the ER, mitochondria, PM, vacuoles, LD, and peroxisomes (Supplementary Table 1). The chosen membrane proteins were genomically tagged with either the N- (VN) or C-terminal fragment (VC) of Venus, a variant of YFP[8] and were checked by colony PCR for the correct insertion. Validation of C' orientation facing the cytosol was verified using complementation to a cytosolic, soluble, Venus half (Supplementary Fig. 2). Strains were imaged using spinning disk microscopy (as above).

**Inducible split PerMit reporter**. To express the PerMit reporter in an inducible manner, Pex25 was genomically tagged in its N' termini using an inducible GAL1 promoter followed by the-VN fragment (VN) of Venus[8], in the background of Tom70-VC strain. Pex25 genomic manipulation was checked by colony PCR. Strains were grown on glucose and were shifted to galactose media at time 0. Strains were imaged using spinning disk microscopy (as above) form time 0 to time 480 min. The signal of the PerMit reporter was visualized starting time ~140 min. No PerMit signal was detected in control cells that were not grown in galactose.

**Split IFP1.4 reporters (split FR)**. The same membranal proteins that were tagged by split Venus were genomically tagged with either the IFP-F[1] or IFP-F[2] halves of the engineered *Deinococcus radiodurans* infrared fluorescent protein IFP1.4[20] and were checked by colony PCR for the correct insertion. Strains were imaged using spinning disk microscopy. To overcome IFP1.4 low-quantum yield and low brightness, the images were taken using high exposure (5.5 s) and the highest laser intensity. Importantly, the FR signals were difficult to detect even with high

exposure and laser intensity. In our hands, adding exogenous Billiverdin (Biliverdin-HCl, Frontier scientific #B14022, 4–100 μg ml$^{-1}$) did not increase the FR signal. This implies that for the low-level interaction that we seek, the FR signal was enabled by the existing endogenous Biliverdin.

**Serial dilutions**. Serial 10-fold dilutions were created by starting with $OD_{600} = 0.1$ of all strains of interest in liquid media and diluting them in 10-fold increments. Cells were then plated using Finnpipette™ F1 Multichannel Pipettes (Thermo Scientific™) on synthetic media supplemented with either 2% glucose or 3% glycerol or 0.2% oleate (+0.1% Tween80) or 2% ethanol agar plates and imaged using Nikon Coolpix P510 digital camera after 2–8 (as indicated) days of growth at 30 °C.

**Measuring PerMit extent in various physiological conditions**. Strains expressing the PerMit reporter and a peroxisomal marker (MS2661 Tom70-VC-His; Pex25-VN-Kan; Pex11-mCherry-Nat) were grown in synthetic media with either 2% glucose or 0.2% oleate (+0.1% Tween80) or 2% ethanol for 4 h and imaged using spinning disk microscopy (see above). The number of Pex11-mCherry and split PerMit puncta per cell were manually scored. $n = 4$, 100 cells per experiment. Data are presented as mean ± s.d.

**Peroxisomes motility**. Peroxisome speed was analyzed using CFP-SKL as a marker. Time-lapse videos at length of 4 min were obtained using a spinning disk microscope (see above). Peroxisomes were segmented and their speed was analyzed using Imaris (see Image analysis). Data represent 25,000 segmented peroxisomes from four or more independent experiments.

**Fzo1 truncation**. Truncation of Fzo1 N′ domains were constructed by tagging the genomic Fzo1 gene with Tef2-mCherry-URA cassette downstream the described domain, creating an N′ truncation of the protein after the indicated nucleotide (547) on the background of an endogenous Fzo1 expressed to allow for normal mitochondrial shape. Primers used are listed here:

FZO1 N′ tag 547Trnc F:
TTATAACAATACTAAAGAAGCACTTCTCAATGCGTTGGATGGTCGACG-GATCCCCGGGTT

FZO1 N′ tag 547Trnc R:
GACCGAGGCCCTGATATTTCGGATATTCGTGTAGCGGAACCTTGTA-CAGCTCGTCCATGC

**Fractionation**. Cells were grown in SD media to exponential phase ($OD_{600} = 0.5$–1), for cell fractionation[74]. Spheroplasts were prepared by treatment with Zymolyase (Zymo Research; Orange, CA). After homogenization of spheroplasts by douncing in cold NMIB (0.6 M sorbitol, 5 mM MgCl$_2$, 50 mM KCl, 100 mM KOAc, 20 mM Hepes pH 7.4) and centrifugation at 3000×$g$, the supernatant (Total fraction) was subjected to centrifugation at 10,170×$g$ for 10 min, yielding a supernatant (Sup) and a mitochondrial enriched pellet fraction (Pellet). Subcellular fractions were assessed for Fzo1 (Anti-Myc tag: 1:1,000 (dilution), 9E10, Invitrogen, R950-25), cytosolic Pgk1 (Anti-Pgk1: 1:20,000 (dilution), Abcam, ab113687), and mitochondrial Por1 proteins (Anti-Por1: 1:1,000 (dilution), Abcam, ab110326). Whole scans of blots are shown in Supplementary Fig. 8.

**Co-immunoprecipitation**. 160 $OD_{600}$ of yeast cells grown in YPD media with 2% (w/v) glucose to the exponential growth phase were disrupted with glass beads (0.4–0.6 μm) in TBS buffer. After centrifugation, the crude membrane fraction was solubilized using 0.2% NG310 for 1 h rotating at 4 °C. HA-Fzo1 was immuno-precipitated using HA-coupled beads (Sigma-Aldrich) for 2 h rotating at 4 °C. Beads were washed 3× with 0.2% NG310 in TBS and HA-Fzo1 was eluted in Laemmli buffer for 20 min shaking at 45 °C. The eluate was split and immuno-decorated with HA-specific (Anti-HA High Affinity, 1:1000 (dilution), Roche, 13565000) and GFP-specific (GFP polyclonal - ChIP Grade, 1:1000 (dilution), Abcam, ab290) antibodies. Whole scans of blots are shown in Supplementary Fig. 9.

**Mitochondrial morphology**. Strains were grown in rich media (YPD) to exponential phase and fixed with 3.7% formaldehyde. Mitochondrial morphology was assessed by following Tom70-GFP (for Supplementary Fig. 5c) or MTS-BFP (for Supplementary Fig. 6d) fluorescence and was characterized as either tubular, fragmented, or globular. Morphology phenotypes were assessed in at least 100 cells. Error bars represent the s.d. from three independent experiments.

**Image analysis**. Image analysis was performed using Imaris v9.02 image analysis software (Bitplane) and the batch analysis extension package. For the peroxisome motility assay, peroxisomes (marked by CFP-SKL) were identified using the Imaris built in spot and tracking functions. Boxplot was created using a matlab script to align a boxplot and the raw data points. The raw data points were split to equal sized bins so the jitter in the $y$ axis is proportional to the relative size of the bin in the population. For the peroxisomes-mitochondria co-localization assays, the organelles identified using the Imaris built in spot function and overlapping

identified organelles were assessed. For calculating the fluorescence average signal area of split MAM or split PerMit the fluorescence signal was identified using the Imaris built in surface function.

**β-oxidation activity measurements**. β-oxidation assays in intact yeast cells[62,75] were performed and optimized for the pH and the amount of protein. Oleate-grown cells were washed in water and resuspended in 0.9% NaCl ($OD_{600} = 1$). Aliquots of 20 μl of cell suspension were used for β-oxidation measurements in 200 μl of 50 mM MES (pH = 6.0) and 0.9% (w/v) NaCl supplemented with 10 μM [1-$^{14}$C]-octanoate. Subsequently, [1-$^{14}$C]-CO$_2$ was trapped with 500 μl 2 M NaOH. The CO$_2$ production and the ASPs were used to quantify the rate of fatty acid oxidation. Results are presented as percentage relative activity to the rate of oxidation of control cells. For Fig. 5c, $n = 5$, for Fig. 5d, $n = 3$.

**Data availability**. Any data that support the findings in this study are available from the authors on reasonable request.

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

## Acknowledgements

Work in the Schuldiner lab was supported by the European Research Council (ERC) (Consolidator grants Peroxisystem 64660), an SFB1190 from the DFG and a DIP (MitoBalance P17516). M.S. is an incumbent of the Dr. Gilbert Omenn and Martha Darling Professorial Chair in Molecular Genetics. The Cohen group was supported by the labex DYNAMO (ANR-11-LABX-0011-DYNAMO). A.H. is funded by NIH grants AG043095 and GM119694. F.R. was supported by Marie Skodowska-Curie ITN (765912), Marie Skodowska-Curie Cofund (713660), ALW Open Program (ALWOP.310) and ZonMW VICI (016.130.606) grants. We thank Ralf Erdmann for kindly sharing with us the CFP-SKL plasmid, Christian Ungermann for the MTS-BFP plasmid, Tomer Ravid for the mCherry-Nat plasmid, Won-Ki Huh for the Split Venus plasmids, Stephen Michnick for the far-red IFP plasmids and discussions, and Christopher Stefan for the Δtether strain. We also thank Noa David Geller and Hanna Vega for

the graphic design; Ebru Erbay for suggesting the PerMit acronym; Maria Bohnert for suggesting the CLIP acronyms; and Inês Gomes Castro for suggesting the LiDER acronym.

## Author contributions

M.S. and E.Z. conceived the study and supervised the experiments. N.S. performed the majority of the experiments. E.Y. identified Fzo1 in peroxisomes and helped N.S. in studying the PerMit contact. N.C. analyzed and quantified the data. C.B. helped N.S. with experimental verification. C..v.R. and L.I. performed the β-oxidation measurements and analyzed the data under the supervision of H.W. and R.W. L.C. and J.M. performed the *Δmdm30* and the mitochondrial morphology experiments under the supervision of M.C. R.S. performed the Fzo1 coIP experiment under the supervision of M.E.H. L.Z. helped to establish the split Venus tool. M.C.M. performed electron microscopy experiments and measurements under the supervision of F.M.R. A.H. has noticed that Fzo1 OE leads to peroxisome clumping on mitochondria. N.S., M.S. and E.Z. wrote the manuscript.

## Additional information

**Competing interests:** The authors declare no competing interests.

