## [Peer Review File · Nature Communications]

Reviewer #1 (Remarks to the Author):

The revised version of this study is improved. It shows how split-Venus can be used to systematically identify contacts sites and now does a better job discussion the caveats of this approach. The split-Venus tether between peroxisomes and mitochondria is used to identify endogenous tethers. One is Fzo1 and evidence is presented that homotypic interactions between Fzo1 in mitochondria and peroxisomes tethers these organelles. The manuscript also suggest that Pex34 is a tether and argues that the Pex34 tether facilitates the transfer of β -oxidation products from peroxisomes to mitochondria. The study is well done and largely convincing. However, many of the findings are over interpreted and the claim that Pex34 affects small molecule transfer is premature. Many of my concerns can be addressed by rewriting but the study would be stronger if some were addressed experimentally, particularly those about Fzo1 localization.

1. The novelty and significance of using split-Venus to identify contacts sites is still overstated. First, it is claimed that there has never been a systematic analysis of contacts (lines 77-78). This ignores Valm et al (ref. 19), which used fluorescence microscopy to systematically look at organelle co-localization. It is true that some colocalization may not indicate the existence of actual contact sites but the same can also be said of detecting potential contacts with irreversible split-Venus. Second, verifying contacts with FR (which is reversible) is only marginally better than using split-Venus. The affinity of the two halves of FR is not known but is likely high enough to promote contact formation. It would be better use a method that does not require even transient organelle tethering for contact visualization. The authors say EM is too much trouble (though others have done it using immuno-gold or APEX to identify organelles) but they could easily quantitate organelle colocalization using fluorescence microscopy. In fact, this was done for one contact in Fig. S3e but could be done for all. At a minimum, the manuscript should have a more balanced discussion of how to verify contact identified with split-Venus.

2. I do not understand why the authors claim that PerMit stabilizes contacts but does not create them. The fact that PerMit is near PDH puncta could indicate that PDH puncta form at contacts after PerMit creates them. There is no way to know whether PerMit (or other split-Venus or split-FP combinations) create contacts without having some independent way to visualize and quantify contacts. The authors may well be right that PerMit does not generate contacts but without better evidence they should have a more balanced discussion.

3. The manuscript states that all reporters of peroxisome-mitochondria contacts gave similar signals (line 160) but there is no quantification of mean fluorescence per puncta, number of puncta per cell, or measurement of the relative abundance of the various fusion proteins used. Either these results should be included or these caveats should be mentioned.

4. The new Fzo1 results are a nice addition. As the authors are surely aware, the idea that Mfn2 is dually localized to the ER and mitochondria and functions as a tether has remained controversial and I have similar concerns about Fzo1. I would be more convinced if there were evidence that Fzo1 is on peroxisomes. The authors suggest that the interaction of Fzo1 with Pex14 and Pex19 indicate it is on peroxisomes but it remains possible that Fzo1 on mitochondria interacts with the proteins. Perhaps fractionation could be used. What percent of Fzo1 is on peroxisomes?

5. The weakest part of this study is the claim that Pex34 affects the transfer of β oxidation products from peroxisomes to mitochondria. I did not mean to suggest that "...a small molecule should not transfer through a contact site." Indeed, the idea that contacts sites facilitate small molecule transfer is attractive but obtaining strong evidence is difficult. This study does not directly measure transfer and instead uses the production rate of CO₂ and Acid Soluble Products (ASP) as indirect indicators of transfer rate. This assumes that: (1) rates of production are primarily determined by β oxidation, (2) that transfer is the rate limiting step of production, and (3) that increased organelle tethering will affect the transfer rate. These may well all be true but there is no evidence to support them. One way to get some indication about the role of contacts in transfer, as I suggested before, would be to artificially separate or tether the organelles and test production rates but, in any case, many more controls are necessary. In the absence of such evidence, the idea that peroxisome-mitochondria tethering affects the transfer rate of β oxidation products is little more than an attractive hypothesis and it remains plausible that Pex34 elimination alters the production of CO₂ and ASP indirectly.

6. Tchekanda et al 2017 says that IFP1.4 fluorescence requires exogenous biliverdin but this is not mentioned in the methods section. How much was used and how long does it take FP to become fluorescent in yeast?

We requested further clarifications from this reviewer regarding the validation of the findings with a method that does not require even transient organelle tethering for contact visualization and on the requirement of controls to claim that Pex34 affects the transfer of β oxidation products from peroxisomes to mitochondria

Currently, the best way to verify membrane contacts is with EM. Verifying all the contacts this study describes with EM is, I think, too much to ask but it is not unreasonable to verify the 4 previously unknown contacts they highlight. Using fluorescence microscopy is second best but is acceptable and, as I said in my review, they have already done this for one of the 4 contacts (Fig. S3e). However, I should also say that the issue of quantitatively measuring contacts and distinguishing real contact from stochastic organelle interactions is a big challenge in the field and part of what I thought the authors should do is simply have a more candid discussion of this. In general, I think any claim of a “new” contact should be verified with a non-invasive method of visualizing contacts.

With regard to the claim that Pex34 affects the transfer of beta-oxidation products, there are 3 ways the authors could go to strengthen the claim.

1. They could provide controls that verify all the assumptions about their findings that I listed in my review. This would be a lot of work.
2. They could determine how lipid metabolism changes in the mutant lacking Pex34. Specifically, they need to determine whether expression of the proteins in the beta-oxidation pathway change in the mutant. They also need to know how pools of free fatty acids change in the mutant and that fatty acid uptake by peroxisomes is not altered.
3. Another possibility would be to show in vitro that tethering peroxisomes to mitochondria alters rates of beta-oxidation.

Reviewer #2 (Remarks to the Author):

In their present manuscript, the authors have addressed some of the concerns expressed by the reviewers, and have given reasons for not addressing most of them.

1. One pervasive concern was the fact that the bimolecular fluorescence complementation constructs induced artificial tethering, blurring most of the claims of the paper. To address this concern, the authors used a revised version of the reporter based on the *Deinococcus radiodurans* bacteriophytochrome, developed by the Michnick lab. This construct induces less artificial tethering

than the original construct, and indeed, when coupled with ER and mitochondria targeting sequences, fails at complementing ERMES deficiency, as the original construct did.

The authors use it to repeat a small subset of experiments, but do not use it pervasively because its fluorescence is too low.

There are a couple of questions about this new construct. First and foremost, the method section indicates that the authors did not add biliverdin as a cofactor, which is indispensable for the reporter to actually work. It is thus unclear how the experiment could actually yield any signal, and suggest that signals may actually be autofluorescence. In addition, the serial dilutions in Supp. figure 1j should also be made in the presence of biliverdin because the cofactor would be necessary for the reporter to perform any kind of tethering. Second, the quantification in Supp. Figure 3e shows an increase of mito-peroxisome co-localisation when the original construct is used. The expectation is that the new construct should not cause a similar increase. This should be tested (again in the presence of biliverdin).

2. Another pervasive concern is that all experiments involve artifact-prone overexpression. It is regrettable that the authors find an excuse by saying that loss-of-function does not work. They should instead find clever ways to show a tethering role for their proteins without resorting to overexpression. This concern is sufficiently important that it was voiced in all three reviews. More experiments are needed to ascertain a tethering role for the proteins when expressed at endogenous levels. For instance, it is still unclear if Fzo1 expressed at endogenous level is substantially found in peroxisomes. There may be clever ways to show that the loop between Fzo1 TM domains is exposed to the peroxisomal lumen?

3. The authors have not been able to show EM pictures of their contact sites, although specifically asked by all three reviewers. They cite the technical difficulty as an excuse. Several groups have used EM to image peroxisomes in *S. cerevisiae*, and this usually works fine in oleic acid-grown cells. For instance, cells may be post-stained for catalase activity, using diaminobenzidine and hydrogen peroxide. The authors should make an effort in that direction to have an independent confirmation showing that their constructs behave as advertised. In the absence of any independent approach, the authors are doomed to circular reasoning (there is a contact because our constructs go there – our construct works because they show us contacts).

As a matter of fact, EM has been suggested by all three reviewers probably because it is the most straightforward thing to do. If truly impossible, the authors should find an alternative way to provide independent proof of their claims.

Other points:

1-Line 92: "It was previously demonstrated that deletions of a single set of tethers most often do not completely abolish a contact site^{11,12,15}."

None of the three references cited can be used in support of this sentence since none of these papers show that contact sites remain in the absence of their respective tethers.

2-Line 120: "For example, the shape of the vacuole-mitochondria (vCLAMP)¹³ and the PM-mitochondria contact (MECA)^{20,21} were crescent-like".

Both references cited actually describe MECAs as point-like structure, and not crescent-like (one more evidence of the tethering power of the construct?).

3-Line 187: "...the ATP binding cassette (ABC) transporter 1 (ABCD1), located to the peroxisomal membrane, is a peroxisome-mitochondria tether protein whose loss of function causes X-linked adrenoleukodystrophy (X-ALD)"

Maybe better as: "...the ATP binding cassette (ABC) transporter 1 (ABCD1), located to the peroxisomal membrane and whose loss of function causes X-linked adrenoleukodystrophy (X-ALD), is a peroxisome-mitochondria tether" (the tether is suggested but the disease causality is confirmed).

4-While the authors have quantified some of their microscopy data, other still need be quantified. In particular, the claim that Fzo1 is not homogeneous but enriched at contact sites needs to be assessed quantitatively and impartially.

Moreover, since overexpressed Fzo1 appears to be both at mitochondria and peroxisomes (Fig. 4D), it will be important to distinguish whether any increased Fzo1 signal at sites of Mito-peroxisome overlap is not merely due to the addition of the mitochondrial and peroxisomal Fzo1 signals.

5-In their response to reviewer #1 (point 6), the authors state that it is not possible to image Fzo1 without overexpressing it. That is not correct. Fzo1 can be N- or C-terminally GFP-tagged at its endogenous locus, under its endogenous promoter (see e.g. Akema et al. 2014) This is an important experiment that should be performed.

6-In their response to point 8 of the same reviewer, the authors state that a metabolic characterization is beyond the scope of this manuscript. If they are not prepared to perform such characterization, then they should consider removing this whole part from the manuscript. I agree with the referee that the analysis is, at this point, too preliminary and that the effects observed here can be explained without a tethering model.

7-In their response to reviewer #3 (point 3), the authors claim that they made an effort to not overexpose or overprocess their micrographs. Yet, most if not all micrographs still appear supersaturated, e.g. the authors should make sure that 1. The background is not suppressed to zero 2. The interesting signal is not saturated.

Reviewer #3 (Remarks to the Author):

This is a revision of a manuscript that was previously submitted to Nature. My first major concern was that the split venus system not only monitors but rather artificially creates membrane contact sites. The authors have added additional control experiments and altered the text to describe more carefully what can be achieved with this system and what its limitations are. This part of the manuscript has been very much improved.

My second major concern (also mentioned by reviewer #2) was that the PerMit contact site was characterized with overexpressed Fzo1 and Pex34. The authors addressed this point to a certain extent for Pex34, but not for Fzo1. As some proteins are known to be shared by mitochondria and peroxisomes, I consider the problem of dual localization or possible mis-localization really important. The experiment shown for Pex34 in the point-by-point response should be described in detail and shown in the paper. The authors write in the abstract that they “uncovered and characterized two tether proteins: Fzo1 and Pex1.” This is a bold statement. Unfortunately, direct evidence supporting a role of endogenous Fzo1 as a PerMit tether protein is still lacking. If the authors cannot provide experiments directly supporting their statement they should tone down their conclusions.

Several references in the list of plasmids are not correct. This should be checked carefully and corrected.

Point by point response to reviewers concerns - NCOMMS-17-33511-T

Reviewer #1

1. *The novelty and significance of using split-Venus to identify contacts sites is still overstated. First, it is claimed that there has never been a systematic analysis of contacts (lines 77-78). This ignores Valm et al (ref. 19), which used fluorescence microscopy to systematically look at organelle co-localization. It is true that some colocalization may not indicate the existence of actual contact sites but the same can also be said of detecting potential contacts with irreversible split-Venus.*

We very much appreciate the work done by *Valm et al* and hence we discuss this paper in our work. We now tone down our claims of novelty and again cite the *Valm et al* manuscript as having identified that all organelles are in proximity to each other.

2. *Second, verifying contacts with FR (which is reversible) is only marginally better than using split-Venus. The affinity of the two halves of FR is not known but is likely high enough to promote contact formation. It would be better use a method that does not require even transient organelle tethering for contact visualization. The authors say EM is too much trouble (though others have done it using immuno-gold or APEX to identify organelles) but they could easily quantitate organelle colocalization using fluorescence microscopy. In fact, this was done for one contact in Fig. S3e but could be done for all. At a minimum, the manuscript should have a more balanced discussion of how to verify contact identified with split-Venus. The affinity of the FR is indeed not known but it is well documented that, unlike the split Venus, it is completely reversible and dissociates immediately upon dissociation of its binding proteins (*Tchekanda et al, Nature Methods 2014*). Hence this protein cannot, in as of itself, support an interaction without being fused to two interacting proteins or membranes. This is a very strong proof that the contacts that we have described are not formed by the intrinsic affinities between two halves of a split reporter. We added a more balanced discussion suggesting which steps could be further done to prove that organelle proximities first observed using the split Venus reporters, and then the split FR reporter, are indeed contact sites. Since the yeast cell is so small it is obvious that all organelles will be proximal to each other at some point hence we feel that performing the fluorescent microscopy experiment will just be uninformative and will not clarify this point.*

3. *I do not understand why the authors claim that PerMit stabilizes contacts but does not create them. The fact that PerMit is near near PDH puncta could indicate that PDH puncta form at contacts after PerMit creates them. There is no way to know whether PerMit (or other split-Venus or split-FP combinations) create contacts without having some independent way to visualize and quantify contacts. The authors may well be right that PerMit does not generate contacts but without better evidence they should have a more balanced discussion.*

The reviewer may have missed an experiment that we added during our previous revision in which we show (Figure 2e) that Pda1, which is part of the PDH complex, is localized to puncta before the PerMit reporter is expressed. Moreover, upon induction of the PerMit reporter expression, signals appear only in places that were previously marked by PDH puncta. This, in our eyes, is clear evidence that the split reporter does not create a new contact site but rather gets recruited to the existing ones. We now better explain the experiment in the text.

4. *The manuscript states that all reporters of peroxisome-mitochondria contacts gave similar signals (line 160) but there is no quantification of mean fluorescence per puncta, number of puncta per cell, or measurement of the relative abundance of the various fusion proteins used. Either these results should be included or these caveats should be mentioned.*

We measured the number of split PerMit puncta per cell in the four different peroxisome-mitochondria reporters using the Imaris software (Supplementary Figure 3c). The quantitation nicely shows that there is no difference in the number of puncta when the different reporters are used.

5. *The new Fzo1 results are a nice addition. As the authors are surely aware, the idea that Mfn2 is dually localized to the ER and mitochondria and functions as a tether has remained controversial and I have similar concerns about Fzo1. I would be more convinced if there were evidence that Fzo1 is on peroxisomes. The authors suggest that the interaction of Fzo1 with Pex14 and Pex19 indicate it is on peroxisomes but it remains possible that Fzo1 on mitochondria interacts with the proteins. Perhaps fractionation could be used. What percent of Fzo1 is on peroxisomes?*

In fact, Mfn2 acting as a tether is no longer controversial since additional papers supporting a role for Mfn2 as a tether have since come out. There is only a single paper contesting this role for Mfn2, and this in our eyes, should not alter our view of the very strong proof of Mfn2 being an ER/Mitochondria tether. However, it is true that at this point we have never shown Fzo1 on peroxisomes when not overexpressed and have not proven that homotypic interactions are governing the contact formation. Hence, we added a more balanced discussion that clearly states that we cannot exclude the possibility that mitochondrial Fzo1 interacts with a peroxisomal protein or *vice versa*. Therefore, we now suggest that the Fzo1-mediated tethering is either created by a homotypic interaction OR by binding to another peroxisomal protein that should be identified in the future. To emphasize this, we have included a model in Figure 5e in which we suggest the different possibilities.

6. *The weakest part of this study is the claim that Pex34 affects the transfer of β oxidation products from peroxisomes to mitochondria. I did not mean to suggest that "...a small molecule should not transfer through a contact site." Indeed, the idea that contacts sites facilitate small molecule transfer is attractive but obtaining strong evidence is difficult. This study does not directly measure transfer and instead uses the production rate of CO₂ and Acid Soluble Products (ASP) as indirect indicators of transfer rate. This assumes that: (1) rates of production are primarily determined by β oxidation, (2) that transfer is the rate limiting step of production, and (3) that increased organelle tethering will affect the transfer rate. These may well all be true but there is no evidence to support them. One way to get some indication about the role of contacts in transfer, as I suggested before, would be to artificially separate or tether the organelles and test production rates but, in any case, many more controls are necessary. In the absence of such evidence, the idea that peroxisome-mitochondria tethering affects the transfer rate of β oxidation products is little more than an attractive hypothesis and it remains plausible that Pex34 elimination alters the production of CO₂ and ASP indirectly.*

We agree with this reviewer's comment that there may be other potential explanations for the phenomena that we see. Hence, we strengthen the point that this is currently a hypothesis and that further work should be done to validate it.

7. *Tchekanda et al 2017 says that IFP1.4 fluorescence requires exogenous biliverdin but this is not mentioned in the methods section. How much was used and how long does it take FP to become fluorescent in yeast?*

We performed the FR experiments without or with different concentration of Biliverdin (4ug/ml, 10ug/ml and 100ug/ml). In our hands we did not see any significant change in the FR fluorescence when Biliverdin was added. After extensive discussions with Prof. Stephen Michnick, it seems that even in their hands, Biliverdin was only required to support maximal fluorescence in cases where large amounts of interacting proteins prevail. This may very well be because yeast contain an inherent amount of endogenous Biliverdin which seems to be sufficient for observing the IFP1.4 signal for low levels of interactions. Since we did not see an added value in using Biliverdin in our specific test cases, we have decided to show the results obtained without adding Biliverdin. To avoid further confusion, we now better explain this in the relevant methods section.

Reviewer #2

1. *One pervasive concern was the fact that the bimolecular fluorescence complementation constructs induced artificial tethering, blurring most of the claims of the paper. To address this concern, the authors used a revised version of the reporter based on the *Deinococcus radiodurans* bacteriophytochrome, developed by the Michnick lab. This construct induces less artificial tethering than the original construct, and indeed, when coupled with ER and mitochondria targeting sequences, fails at complementing ERMES deficiency, as the original construct did.*

The authors use it to repeat a small subset of experiments, but do not use it pervasively because its fluorescence is too low.

There are a couple of question about this new construct. First and foremost, the method section indicates that the authors did not add biliverdin as a cofactor, which is indispensable for the reporter to actually work. It is thus unclear how the experiment could actually yield any signal, and suggest that signals may actually be autofluorescence. In addition, the serial dilutions in Supp. figure 1j should also be made in the presence of biliverdin because the cofactor would be necessary for the reporter to perform any kind of tethering.

Second, the quantification in Supp. Figure 3e shows an increase of mito-peroxisome co-localisation when the original construct is used. The expectation is that the new construct should not cause a similar increase. This should be tested (again in the presence of biliverdin).

Please see our response to Reviewer 1 regarding Biliverdin. To the same extent, since no additional fluorescence was seen we do not expect more binding and hence see no point in repeating the growth experiment.

Regarding the quantitation of mitochondria-peroxisome co-localization, unfortunately, the low fluorescence of the FR makes it unsuitable for the type of quantitation possible in Figure 3e.

2. Another pervasive concern is that all experiments involve artifact-prone overexpression. It is regrettable that the authors find an excuse by saying that loss-of-function does not work. They should instead find clever ways to show a tethering role for their proteins without resorting to overexpression. This concern is sufficiently important that it was voiced in all three reviews. More experiments are needed to ascertain a tethering role for the proteins when expressed at endogenous levels. For instance, it is still unclear if Fzo1 expressed at endogenous level is substantially found in peroxisomes. There may be clever ways to show that the loop between Fzo1 TM domains is exposed to the peroxisomal lumen?

We have shown extensively that deleting even multiple tethers in yeast does not cause loss of the contact site. For example – deleting the ERMES complex does not visually alter the extent of the Mito/ER contact as Lam6 takes over as a tether. The ER/PM contact site has been shown to require six pairs of tethers to be deleted before any significant decrease was observed. We have deleted Pex34, Fzo1 and Pex11 – all suggested PerMit tethers and no reduction in contact could be observed. This would suggest additional tethers await discovery.

Since Fzo1 was also observed in an organelle fraction that is not mitochondria and in puncta co-localized to peroxisomes when *MDM30* was deleted, which is not a strong overexpression, we think that this is good proof of its peroxisomal localization. Still, at this stage we agree with the reviewer that we can not exclude the possibility that the peroxisome-mitochondria tethering that is mediated by Fzo1 involves interaction with a peroxisomal protein. Hence, we now mention the two options in our manuscript and emphasize it graphically in our final model figure.

*3. The authors have not been able to show EM pictures of their contact sites, although specifically asked by all three reviewers. They cite the technical difficulty as an excuse. Several groups have used EM to image peroxisomes in *S. cerevisiae*, and this usually works fine in oleic acid-grown cells. For instance, cells may be post-stained for catalase activity, using diamino-benzidine and hydrogen peroxide. The authors should make an effort in that direction to have an independent confirmation showing that their constructs behave as advertised. In the absence of any independent approach, the authors are doomed to circular reasoning (there is a contact because our constructs go there – our construct work because they show us contacts). As a matter of fact, EM has been suggested by all three reviewers probably because it is the most straightforward thing to do. If truly impossible, the authors should find an alternative way to provide independent proof of their claims.*

We have made dramatic effort to show the peroxisome-mitochondria proximity using EM with no success so far. We will not be able to pursue this direction further at this point.

Other points:

1-Line 92: "It was previously demonstrated that deletions of a single set of tethers most often do not completely abolish a contact site^{11,12,15}."

None of the three references cited can be used in support of this sentence since none of these papers show that contact sites remain in the absence of their respective tethers.

We apologize for this mistake and have now updated the references. There must have been a glitch in our reference program during our revision process. We intended to put the below references there:

1. The reference of Elbaz-Alon et al, *Cell Reports* 2015 – Figure 1G shows that the ER/Mito contact site is present even when Lam6, a well-studied tether, is deleted. Figure S1B shows that the ER/mito contact site is present at the same amount and extent when any of the ERMES components are deleted.

2. The reference of Manford et al, *Dev Cell* 2012 which exemplifies the $\Delta tether$ strain showing that only loss of six tethering proteins could reduce the ER/PM contact significantly.
3. The reference of Henne et al, *JCB* 2015 which shows (For example see Figure 4) that in the Nucleus Vacuole Junction contact site, loss of the previous identified tethers Nvj1/Vac8 does not abrogate the contact as Nvj3 still holds it together.

Finally, we also cite our own work (Eisenberg-Bord et al, *Dev Cell* 2016) where we have summarized all known tethers suggested to date for each yeast contact site. This data clearly shows that there are multiple tethers for each contact explaining why a reduction in contact cannot be easily seen by eliminating a single or even two or three tethering molecules. In fact, in the past experiments showing that contact sites can be reduced in size or number by eliminating a single tether may have been skewed by the use of the “partner” protein for the tether as proof that the contact site is dissociating while in fact this only shows that the specific complex is dissociating.

2-Line 120: *"For example, the shape of the vacuole-mitochondria (vCLAMP)13 and the PM-mitochondria contact (MECA)20,21 were crescent-like"*.

Both references cited actually describe MECAs as point-like structure, and not crescent-like (one more evidence of the tethering power of the construct?).

We thank the reviewer for highlighting this point and we now rephrased the relevant sentence. We clearly discuss the tethering power of our construct and its limitations in the paper.

3-Line 187: *"...the ATP binding cassette (ABC) transporter 1 (ABCD1), located to the peroxisomal membrane, is a peroxisome-mitochondria tether protein whose loss of function causes X-linked adrenoleukodystrophy (X-ALD)"*

Maybe better as: "...the ATP binding cassette (ABC) transporter 1 (ABCD1), located to the peroxisomal membrane and whose loss of function causes X-linked adrenoleukodystrophy (X-ALD), is a peroxisome-mitochondria tether" (the tether is suggested but the disease causality is confirmed).

We rephrased the sentence.

4-While the authors have quantified some of their microscopy data, other still need be quantified. In particular, the claim that Fzo1 is not homogeneous but enriched at contact sites needs to be assessed quantitatively and impartially.

Moreover, since overexpressed Fzo1 appears to be both at mitochondria and peroxisomes (Fig. 4D), it will be important to distinguish whether any increased Fzo1 signal at sites of Mito-peroxisome overlap is not merely due to the addition of the mitochondrial and peroxisomal Fzo1 signals.

We have made an effort to quantify the Fzo1 distribution but without success. Hence, we now better explain Figure 4a to make sure that it is clear that Fzo1 is not expressed all over the mitochondria but is enriched in niches in which the PerMit reporter is localized.

5-In their response to reviewer #1 (point 6), the authors state that it is not possible to image Fzo1 without overexpressing it. That is not correct. Fzo1 can be N- or C-terminally GFP-tagged at its endogenous locus, under its endogenous promoter (see e.g. Akema et al. 2014) This is an important experiment that should be performed.

We apologize for not being clear about this point. We meant that it is impossible to perform the experiment using immunofluorescence. Fzo1 can indeed be tagged at both termini. However, when Fzo1-GFP was expressed under its native promoter we could not see it localized to peroxisomes. This may be because no Fzo1 is on peroxisomes – a reason that we are now highlighting in our discussion, or that only very few molecules exist on this tiny organelle making them hard to track.

6-In their response to point 8 of the same reviewer, the authors state that a metabolic characterization is beyond the scope of this manuscript. If they are not prepared to perform such characterization, then they should consider removing this whole part from the manuscript. I agree with the referee that the analysis is, at this point, too preliminary and that the effects observed here can be explained without a tethering model.

We toned down the message of this section as suggested by both reviewers.

7-In their response to reviewer #3 (point 3), the authors claim that they made an effort to not overexpose or overprocess their micrographs. Yet, most if not all micrographs still appear supersaturated, e.g. the authors should make sure that 1. The background is not suppressed to zero 2. The interesting signal is not saturated.

We created the micrographs by reducing the auto fluorescence for each experiment using a control-non fluorescent strain. Removing auto fluorescence gives a clear signal that is better visualized when using multi-color images, obviously without manipulating the data. We leave the decision to the editor (we attached two versions of Figure 2 for comparison. Left – previous version in which the auto fluorescence is eliminated).

Reviewer #3

1. My second major concern (also mentioned by reviewer #2) was that the PerMit contact site was characterized with overexpressed *Fzo1* and *Pex34*. The authors addressed this point to a certain extent for *Pex34*, but not for *Fzo1*. As some proteins are known to be shared by mitochondria and peroxisomes, I consider the problem of dual localization or possible mis-localization really important. The experiment shown for *Pex34* in the point-by-point response should be described in detail and shown in the paper. The authors write in the abstract that they “uncovered and characterized two tether proteins: *Fzo1* and *Pex1*.” This is a bold statement. Unfortunately, direct evidence supporting a role of endogenous *Fzo1* as a PerMit tether protein is still lacking. If the authors cannot provide experiments directly supporting their statement they should tone down their conclusions.

While we believe that the support we provide of the two new tethering molecules is *en par* with all other literature out there on tethers, we have now toned down our message. We also added the figure that we put in the response to reviewers concerns the previous round as Supplementary Figure 6i:

2. *Several references in the list of plasmids are not correct. This should be checked carefully and corrected. We apologize for this mistake and thank the reviewer for catching this glitch of the reference program. We have now fixed the mistake.*